# Delineation of Grade II and III Gliomas Investigated by 7T MRI: An Inter-Observer Pilot Study

**DOI:** 10.3390/diagnostics13081365

**Published:** 2023-04-07

**Authors:** Martin Prener, Giske Opheim, Helle Juhl Simonsen, Christina Malling Engelmann, Morten Ziebell, Jonathan Carlsen, Olaf B. Paulson

**Affiliations:** 1Neurobiology Research Unit, Rigshospitalet Blegdamsvej, 2100 Copenhagen, Denmark; martin.prener@nru.dk (M.P.); gopheim@gmail.com (G.O.); 2Functional Imaging Unit, Department of Clinical Physiology, Nuclear Medicine and PET, Rigshospitalet Glostrup, 2600 Copenhagen, Denmark; helle.juhl.simonsen@regionh.dk; 3Department of Neurosurgery, Rigshospitalet Blegdamsvej, 2100 Copenhagen, Denmark; x80q@kk.dk (C.M.E.); mzi@regionsjaelland.dk (M.Z.); 4Department of Radiology, Rigshospitalet Blegdamsvej, 2100 Copenhagen, Denmark; jonathan.frederik.carlsen@regionh.dk; 5Faculty of Health and Medical Sciences, University of Copenhagen, 2200 Copenhagen, Denmark

**Keywords:** low-grade gliomas, MRI, ultra-high field, volumetric analysis, delineation of tumor border

## Abstract

Purpose: Diffuse low-grade gliomas (DLGGs) are low-malignancy brain tumors originating from the glial cells of the brain growing continuously and infiltratively along the neural axons and infiltrating the surrounding brain tissue. DLGGs usually transform into higher malignancy, causing progressive disability and premature death. MRI scans are valuable when assessing soft tissue abnormalities, but, due to the infiltrative properties of DLGGs, delineating the tumor borders is a challenging task. Therefore, the aim of this study was to explore the difference in gross tumor volume (GTV) of DLGGs delineated from 7 Tesla and 3 Tesla MRI scans. Method: Patients were recruited at the department of neurosurgery and were scanned in both a 7T and a 3T MRI scanner prior to the operation. Two observers delineated the tumors using semi-automatic delineation software. The results from each observer were blinded to the other observer’s delineation. Results: Comparing GTVs from 7T and 3T, the percentage difference varied up to 40.4% on the T2-weighted images. The percentage difference in GTV varied up to 15.3% on the fluid-attenuated inversion recovery (FLAIR) images. On the T2-weighted images, most cases varied by approximately 15%; on the FLAIR sequence, half of the cases varied by approximately 5% and the other half by approximately 15%. The overall inter-observer agreement was near perfect, with an intraclass correlation of 0.969. The intraclass correlation was better on the FLAIR sequence than on the T2 sequence. Conclusion: Overall, the GTVs delineated from 7T images were smaller. The increase in field strength improved the inter-observer agreement only on the FLAIR sequence.

## 1. Introduction

World Health Organization (WHO) grade II gliomas, also known as diffuse low-grade gliomas (DLGGs), are brain tumors that originate from astrocytes or oligodendrocytes [1,2]. The classification of the tumors is based on histopathological analysis, and, since 2016, genetic and molecular parameters [3]. In 2021 the WHO updated this classification [4]. The parameters used to distinguish between grade II and III are cellular proliferation and early vascular changes [5,6]. WHO grade II DLGGs can be divided into two subgroups: (1) oligodendrogliomas, which are defined by a 1p/19q codeletion (loss of short arm of chromosome 1 and loss of long arm of chromosome 19 and isocitrate dehydrogenase (IDH)-mutation), and (2) astrocytomas, which are defined by the IDH-mutation. Furthermore, WHO grade II tumors are considered benign, which is misleading considering their infiltrative properties. DLGG is a progressive, invasive, chronic disease of the central nervous system with limited possibilities for treatment that eventually leads to a shorter life expectancy [7]. The subtypes of DLGGs differ substantially in overall survival. When surgically treated, the median overall survival of DLGGs is approximately 15 years. For oligodendrogliomas, it is statistically higher than for astrocytomas [5,7,8]. DLGGs usually transform to a higher level of malignancy, which enhances disease development and eventually leads to premature death. There seems to be a survival benefit of early resection rather than a watchful waiting [8]. The diffuse infiltration seen in DLGGs does not destroy the surrounding tissue. Contrarily, the surrounding tissue is functional to a large extent with the presence of only sporadic tumor cells. DLGG cells have been found up to 20 mm from magnetic resonance imaging (MRI) abnormalities [9]. Thus, although MRI is valuable when assessing soft tissue abnormalities, delineation of the tumor periphery and peritumoral areas is a challenging task. MRI at current clinically used field strengths might pose a limitation with regard to revealing signal changes due to diffuse microscopic pathology. Structural 7T MRI was able to detect pituitary microadenomas in 93.75% whereas for 3T it was 75% in relation to intraoperative findings in a cohort of 16 patients [10]. A study conducted with 10 neuroradiologists found that the 7T structural scans compared to the 3T structural scans were more conspicuous in a variety of neurological diseases including neoplasm [11].

### Delineation of DLGGs on 7 Tesla MR Images

We hypothesize that the increase in field strength at 7T will refine the tumor border, but we have no hypothesis regarding gross tumor volume (GTV) being larger or smaller. The higher signal-to-noise ratio (SNR) at 7T allows for shorter scan time with the same resolution as conventional field strengths and/or higher resolution within the same clinical scan time [12]. One study compared 3T and 7T FLAIR images in patients with glioblastomas, which showed the GTV was 7.4% smaller on 7T compared to 3T [13]. However, to our knowledge, no studies have addressed the potential clinical yield from 7T MRI in patients with DLGGs. Therefore, in the present pilot study, we aimed to investigate the difference in GTV of DLGGs delineated from clinical 3T and 7T MRI scans.

## 2. Methods

### 2.1. Patient Population

During the study period, seven patients were scanned at both 3T and 7T. Patients were recruited from the Neurosurgical Department, Rigshospitalet, Copenhagen. Several patients had to be excluded due to tattoos within the set safety margins of <30 cm from the coil. Further, the transportation between Rigshospitalet and Hvidovre Hospital, where the 7T scanner was located, made it difficult to recruit patients. For these reasons we were only able to recruit seven patients within two years. None of the patients had received any treatment prior to the scans. The project was approved by the Ethical Committee of the Capital Region of Denmark (H-4-2014-134_3), and all participants signed consent forms prior to scanning, which included consent for publication. Further, as the 7T scanner was not CE-marked, the study was also approved by the Danish Medicines Agency (2016101738, 2017081122). The data processing was approved by the Danish Data Inspectorate (RH-2016-387, I-Suite nr. 05172).

### 2.2. Comments on Histopathology of Patients

The histopathological analyses showed that three of the seven patients had WHO grade III and not grade II, as assumed when the patients were enrolled in the study. As stated in the introduction, the WHO glioma classification uses mitotic activity to differentiate between grade II and grade III. It seems that WHO grades II and III gliomas have no significant difference in survival, in IDH-mutant, or any difference in age of presentation [5,6]. Therefore, we included IDH-mutant grade III in our material. There is substantial inter-observer variability among neuropathologists when differentiating between grade II and grade III gliomas [14]. Furthermore, MRI classically differentiates between low-grade and high-grade glioma based on whether the tumor contrast enhances or not. However, a study with 927 histologically proven WHO grade II gliomas showed that 143 (15.9%) had contrast enhancement despite being low-grade [15]. A different study with 314 histologically proven gliomas, 243 high-grade and 71 low-grade, showed that the probability that a contrast-enhanced tumor in high-grade was 86,7%, and 37% of the non-enhancing gliomas were high-grade. On the other hand, 46.5% of the low-grade gliomas were enhanced [16]. This further emphasizes the overlap between grade II and grade III gliomas. 

### 2.3. MRI Protocol

Each patient underwent 3T and 7T no more than a month apart to ensure minimal tumor growth in the meantime. The 3T MRI system was a Siemens MAGNETOM Prisma Fit 3T, Syngo MR D13D SP01 scanner with a Siemens 64-channel head coil. The scanner was installed at the Department of Radiology, Copenhagen University Hospital Rigshospitalet.

The 7T MR system was an actively shielded Philips Achieva 7T MR system (Philips Healthcare, Best, The Netherlands) equipped with a two-channel volume transmit head coil with a 32-channel receiver array (Nova Medical Inc., Burlington, MA, USA). The scanner was installed at the Danish Research Centre for Magnetic Resonance, Copenhagen University Hospital Hvidovre. We used a 3D-based B_1_^+^ scaling, which greatly reduced the spatially varying inhomogeneities [17]. On top of the B_1_^+^ scaling, we applied dielectric pads (size 19 × 19 cm, Multiwave Technologies Ltd., Geneva, Switzerland) on both sides of the head to further minimize B_1_^+^-induced inhomogeneities. We also used interleaved fat-navigator-based prospective motion correction for the structural sequences [18], which reduced the need for re-scans and, thus, patient discomfort caused by prolonged scan duration.

Scanning protocol settings for 3T were as follows: 3D turbo spin-echo (TSE)-T2 (0.4 × 0.4 × 0.9 mm) (TE = 117) and 3D T2 FLAIR (0.4 × 0.4 × 0.9 mm) (TE = 95). Scanning protocol settings for 7T were as follows: 3D TSE-T2 (0.7 × 0.7 × 0.7 mm) (TE = 60) and 3D FLAIR (0.7 × 0.7 × 0.7 mm) (TE = 347). Scan time for 3T was as follows: 3D TSE-T2 was 5 m 10 s and 3D T2 FLAIR was 5 m 40 s. Scan time for 7T was as follows: 3D TSE-T2 was 10 m 30 s and 3D FLAIR was 7 m 30 s. 3T T2: TE = 117. 

We calculated the signal-to-noise ratio (SNR) and its standard deviation (SD) for both grey and white matter and the contrast-to-noise ratio (CNR) between grey and white matter. We used an average of two reference points outside of the patient’s head approximately 1 cm from the patient. For 3T T2 the mean SNR_grey_ was 237 ± 114, the mean SNR_white_ was 178 ± 82, and CNR was 1.34 ± 0.15. For 3T FLAIR the mean SNR_grey_ was 239 ± 101, the mean SNR_white_ was 206 ± 90, and CNR was 1.19 ± 0.11. For 7T T2 the mean SNR_grey_ was 287 ± 181, the mean SNR_white_ was 175 ± 99, and CNR was 1.60 ± 0.21. For 7T FLAIR the mean SNR_grey_ was 836 ± 342, the mean SNR_white_ was 515 ± 272, and CNR was 1.70 ± 0.28. 

Patients were screened twice for MRI contraindications: once prior to the 3T scan and once prior to the 7T scan. The 7T MR safety protocols are currently formed locally at each 7T MR center around the world, which is different from the 3T MR safety protocol where general guidelines are accepted internationally.

### 2.4. Analysis

This study was set up as an inter-observer study in which two observers with different clinical backgrounds (medical student/newly educated physician, M.P., (observer 1) and neuroradiologist, J.C., (observer 2)) performed semi-automatic delineations of GTV using Mirada (Mirada Medical Ltd., Oxford, UK). The tumor border was defined visually. Each observer performed the volume delineation once. There were no hurdles while handling the 7T data using the Mirada software. To minimize bias, the two observers were blinded to the results of the other observer until all data sets were acquired and the delineations completed. The order of the structural MR images in which delineations were performed was randomized for observer 2. Observer 1 performed the delineations as soon as data were available. An example of delineations on both 3T and 7T MR images 3D TSE T2 is shown in Figure 1. The difference in GTV from 3T to 7T was calculated in percentage as (GTV_3T_ − GTV_7T_)/GTV_3T_. It was categorized into three groups: larger, same, and smaller. The same size was defined at ±5%. The data were further stratified into subgroups such as WHO grade, size (“Big” and “Small” tumor with a cut-off of 15 cm^3^), and histopathology. Further, we calculated the intraclass correlation coefficient (ICC), which is a measure of reliability between different observers expressed as a number between 0 and 1, with 0 meaning zero reliability and 1 meaning perfect reliability. The average volume size and the average difference between the GTV delineated by the two observers were also calculated both as absolutes and in relative units as (GTV_obs1_ − GTV_obs2_)/GTV_obs2_. 

## 3. Results 

Of the seven patients included in the study, six were operated on while the remainder waited under watchful observation. One of the operated patients had an oligodendroglioma and five an astrocytoma. One of the astrocytoma patients had both astrocytoma and oligodendroglioma characteristics. IDH mutation was present in all six. The tumor location was supratentorial in all cases (frontal, frontotemporal, temporal, insula, parietal, or occipital in the operated and frontal in the non-operated).

The data from the delineations from both observers on the T2-weighted images are shown in Table 1. For observer 1, the GTVs were overall smaller on 7T than on 3T, with only one GTV being larger on 7T than on 3T. The percentage difference varied between −1.6 and 17.5 for observer 1. For observer 2, half of the GTV were larger on 7T than on 3T, and half were smaller on 7T than on 3T. The percentage difference varied between −34.4 and 40.4 for observer 2.

The data from the delineations from both observer 1 and observer 2 on FLAIR images are shown in Table 2. For observer 1, the GTVs were overall smaller on 7T than on 3T, with only one GTV being larger on 7T than on 3T. The percentage difference varied between −3.4 and 15.3 for observer 1. For observer 2, the GTVs were overall smaller on 7T than on 3T, with only one GTV being larger on 7T than on 3T. The percentage difference varied between −5.4 and 14.6 for observer 2.

For observer 1, all tumors but one had GTVs smaller on 7T compared to 3T on T2, the last one was similar in size. For the FLAIR scans the GTVs for three were smaller on 7T, and the other four were of the same size. For observer 2, the T2 scans showed that two of the seven GTVs were larger on 7T than on 3T, one was the same size, and four were smaller on 7T than on 3T. For observer 2, the FLAIR images revealed only one as larger on 7T, one of the same size, and five smaller on 7T, see Appendix A. When all the patients were included, the majority of the GTVs were smaller on 7T than on 3T, with the exception of observer 1 on the FLAIR sequence, where four of them were approximately the same size; see Appendix A.

The categories “Big” tumor and “Small” tumor, with a cut-off of 15 cm^3^, are very similar in their distribution, with both primarily being smaller on 7T than on 3T, with the exception of the FLAIR sequence for observer 1. The same pattern is seen when separating GTV differences by WHO grade. The distribution between oligodendroglioma and astrocytoma is difficult to analyze due to the limited number of oligodendroglioma patients.

The two observers only strongly disagreed with one of the GTVs on the T2 sequence, as observer 1 found that the GTV was larger on 3T than on 7T, and observer 2 found the opposite. Seven of the remaining thirteen delineations showed a slight disagreement between the two observers, meaning that one found it to be the same size, while the other thought they were either larger or smaller. The remaining six showed good agreement among the observers. 

The ICC varied between 0.969 and 0.989, which is near perfect, as shown in Table 3. The average volume was 24.1 cm3, and the GTV delineated from T2 is smaller than on FLAIR; the GTV is smaller on 7T than on 3T. The smallest volume is delineated from the T2 sequence using the ultra-high field MRI. The highest ICC was 0.989 for FLAIR on the 7T, and it has the lowest mean difference between observers in percentage. 

The difference in GTVs between the two observers in percentage as a function of the average GTV at each field strength and MRI sequence is shown in Figure 2. The scatterplots from Figure 2A–C show an even distribution of the dots between negative and positive percentage deviations. Figure 2D shows that six out of seven dots have a positive y-value. 

## 4. Discussion

The main findings were that tumor delineation with 3T and 7T MR were comparable, although tumor volumes at 7T were slightly smaller than at 3T, especially for T2-weighted images. In addition, the inter-observer agreement was near perfect. In the clinical setting, this indicates that a good estimation of tumor volume can be well obtained even with 7T and 3T MR in most cases of DLGGs. 

### 4.1. Delineation of Tumor Borders and Infiltration

The data showed a trend towards smaller GTVs on 7T compared to 3T, which corresponds with a previous study done on glioblastomas [13]. The SNR on FLAIR 7T compared to 3T was approximately 2.5 times bigger, which corresponds well with previous studies [19]. 

This finding is interesting considering that the tumor and surrounding tissue can be divided into three different areas with merging borders: the tumor, infiltrated but otherwise normal tissue, and normal brain tissue. These borders, however, are not well defined, neither histologically nor radiologically, because DLGGs are not homogenous tumor masses. Using antibodies as a marker, it has been estimated that between 15 and 30% of the tumor consists of non-neoplastic cells, primarily microglia cells, and macrophages [20,21], which further emphasizes its diffuse nature. Histological tumor borders and radiological tumor borders are not perfectly aligned, and the golden standard of tumor delineation is histological. One study looked into the difference between the radiological tumor border on T2-weighted and FLAIR images and the histological tumor border using IDH-specific antibodies on the resected tissue [14]. It showed that tumor cells were found outside the delineation on 3T T2-weighted and FLAIR images, and also that radiologically defined tumor tissue was found outside the histologically defined tumor, which complicates the relationship between the two even more. Since the radiologically defined tumor and histologically defined tumor border do not correlate fully, as mentioned, [9] one might expect that 7T with its higher SNR might give a better correlation to the histologically defined tumor borders. Further studies comparing the 7T delineated tumor border with the histologically defined tumor border are needed to answer that question.

It is also well-known, that tumor delineation on MRI scans is dependent on window width and level, which was not accounted for in this study [22]. Attempts have been made to standardize MRI presentation, but no consensus on how to best display MRI scans has been achieved [23].

Another study compared MRI abnormalities with stereotactic biopsies obtained from eight untreated oligodendroglioma patients and found that MRI abnormalities correlated with the edema fraction and not with the total cell concentration, tumor cell concentration, or cycling tumor cell concentration [24]. The signal threshold for their T2-weighted sequence was thus approximately an edema fraction of 20%. The authors did not specify the field strength of their MRI scans. 

We found no pattern when stratifying the data into subgroups such as WHO grade, size (“Big” and “Small” tumor with a cut-off of 15 cm^3^), and histopathology. This was likely due to the small sample size.

### 4.2. Inter-Observer Agreement

Overall, the inter-observer reliability was near perfect with ICC scores varying from 0.969 to 0.989. Observer 1 (unexperienced observer) had the tendency to overestimate the GTV compared to observer 2 (neuroradiologist) on the FLAIR sequence at 7T. Despite the small number of scans, the data indicate a minor non-biased difference in GTV between the two observers. The observer agreement was better at 7T than on 3T for the FLAIR scan but not the T2-weighted scan. The increase in field strength thus improved the inter-observer agreement. 

Common for both observers was that the difference between GTV from 3T to 7T varied more on T2 than on FLAIR. The very best ICC was obtained with the FLAIR images at 7T. This might indicate that FLAIR is superior for the delineation of DLGG tumor borders. In any instance, there was excellent agreement between the two observers both at the 3T and 7T. The differences are bigger for observer 2 than for observer 1. Observer 2 is a neuroradiologist, thus it is reasonable to assume that the precision of the delineation is at a higher standard. Observer 2 did the delineation in random order. Observer 1, who also conducted the scan, performed the delineation as soon as the data was available. This means that observer 1 was biased to prior delineations to a much larger degree than observer 2. 

### 4.3. Differences and Challenges in Clinical 7T MRI

The challenges of clinical 7T MRI are not different from the challenges of 3T MRI, they are merely amplified [12]. The increased field strength of 7 Tesla poses certain challenges such as increased inhomogeneity of the applied transmit field (B_1_). Particularly the anterior- and posterior-inferior areas are affected, with variations introduced by head size. This can, to some extent, be mitigated by using dielectric pads, although it is still not optimal, neither in the present classical hardware set-up nor in multi-transmit 7T systems. 

Besides the B1 inhomogeneity, the image contrast differs between 3T and 7T. The contrast of the 7T scans can be modified, but it will probably never be identical to the contrast of the 3T scans. This difference in contrast from a healthy brain to a tumor might add to the difference in delineated GTVs from 7T to. 3T. The smaller GTVs on 7T might be due to the better discrimination of tumor tissue from healthy tissue on 7T compared to 3T. Furthermore, the problem regarding partial volume is smaller on 7T due to the smaller voxel sizes. 

Observer 2 is used to analyzing conventional field strength MRI scans, which means that he might have a bias of habit, which a new observer will not have. The new observer might be more adaptive than more experienced observers, which was the rationale behind establishing the multi-observer study design. 

It should further be noted that the increased specific absorption rate (SAR) at 7T, which describes the heating of the scanned tissue, is not a unique problem for ultra-high field MRI. SAR heating limits 3T scanners as well. The global SAR value, which is the mean SAR value over a certain volume, increases with the square of the applied field strength. The SAR constraints are always a limiting factor. Of note, the SAR monitor in both systems is set with very large margins, and SAR heating has not been a problem in our project.

The 7T MRI scanner used in this project was not CE-marked, as most 7T scanners across vendors, which is the reason for the lack of global 7T MR safety consensus. Each 7T center, therefore, establishes its own safety rules, which means that the three patients excluded based on their tattoos at our center might have been included in other centers that, e.g., assert the field-of-view borders at the range where tattoos are considered prohibited. These issues limit the clinical use of the 7T MRI scanner, but within the 7T MR community, the global 7T MRI safety consensus is gaining increased focus. A 7T MRI safety consensus will very likely help any 7T MR center when evaluating the inclusion of patients.

## 5. Conclusions

The clinical benefits of structural 7T compared to 3T have shown a better delineation of structures in several studies. In general, the delineations of DLGG in the present pilot study yielded smaller GTVs from 7T images compared to 3T images, which corresponds with studies conducted on glioblastoma [13]. Overall, there was an excellent agreement between the inexperienced delineator and the experienced. The difference between GTVs 7T to 3T varied more on T2 than on FLAIR. The inter-observer variability was also smaller on FLAIR than on the T2-weighted images. This study further indicates that a neuroradiologist can calmly entrust an inexperienced, but dedicated co-worker to delineate grade 2 and 3 non-operated gliomas on MRI scans. The overall conclusion is that a good estimation of tumor volume can be obtained with both 3T and 7T MRI.

## Figures and Tables

**Figure 1 diagnostics-13-01365-f001:**
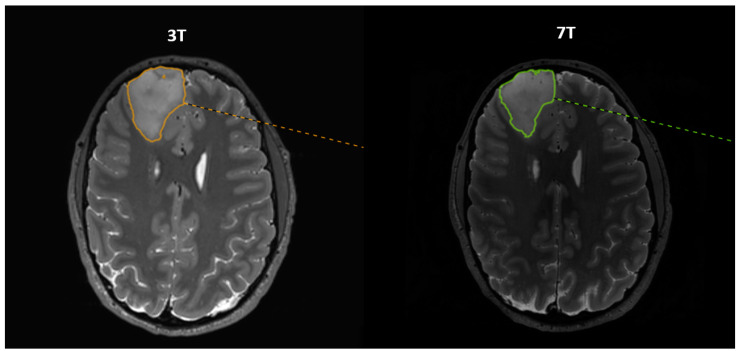
Corresponding axial slices from 3T and 7T 3D T2 images with delineation of a frontal tumor. The patient was a 35-year-old male with DLGG grade III situated in the right frontal lobe. The slices depict the largest cross-section of the tumor. Of note, the 3T and 7T slices are not completely identical due to slightly differing angulations during the reconstruction of axial images.

**Figure 2 diagnostics-13-01365-f002:**
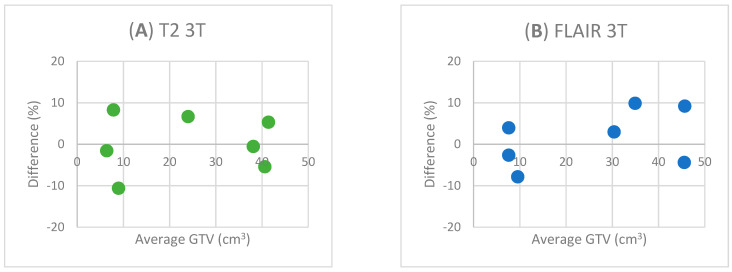
The difference in GTV between the two observers is shown as a percentage. On the *x*-axis, the average GTV delineated from each field strength and sequence is shown. On the *y*-axis, a percentage is shown. The difference between the observers is defined as (GTV_obs1_ − GTV_obs2_)/(GTV_obs1_ + GTV_obs2_).

**Table 1 diagnostics-13-01365-t001:** Gross tumor volumes (GTV) from 3D TSE T2 MR images.

	Observer 1			Observer 2		
Pt. No.	GTV cm^3^ 3T	GTV cm^3^ 7T	GTV cm^3 ^ Difference (%) *	GTV cm^3^ 3T	GTV cm^3^ 7T	GTV cm^3^ Difference (%) *
**1**	25.6	22.0	3.6 (14.1)	22.4	30.1	−7.7 (−34.4)
**2**	38.4	32.3	6.1 (15.9)	42.8	40.5	2.3 (5.4)
**3**	6.3	6.4	−0.1 (−1.6)	6.5	7.2	−0.7 (−10.8)
**4**	8.0	6.6	1.4 (17.5)	9.9	5.9	4.0 (40.4)
**5**	43.6	37.7	5.9 (15.6)	39.2	40.6	−1.4 (−3.6)
**6**	37.9	34.2	3.7 (10.8)	38.3	34.3	4.0 (10.4)
**7**	8.5	7.8	0.7 (9.0)	7.2	6.6	0.6 (8.3)

* (GTV_3T_ − GTV_7T_)/GTV_3T_.

**Table 2 diagnostics-13-01365-t002:** Gross tumor volumes (GTV) from 3D FLAIR MR images.

	Observer 1			Observer 2		
Pt. No.	GTV cm^3^ 3T	GTV cm^3^ 7T	GTV cm^3^ Difference (%) *	GTV cm^3^ 3T	GTV cm^3^ 7T	GTV cm^3^ Difference (%) *
**1**	31.3	26.5	4.8 (15.3)	29.5	25.7	3.8 (12.9)
**2**	43.6	39.0	4.6 (10.6)	47.6	40.7	6.9 (14.5)
**3**	7.9	7.7	0.2 (2.5)	7.3	6.8	0.5 (6.8)
**4**	8.8	9.1	−0.3 (−3.4)	10.3	8.8	1.5 (14.6)
**5**	49.9	47.2	2.7 (5.7)	41.5	41.2	0.3 (0.7)
**6**	38.4	38.0	0.4 (1.1)	31.5	33.2	−1.7 (−5.4)
**7**	7.4	7.6	0.2 (2.6)	7.8	6.7	1.1 (14.1)

* (GTV_3T_ − GTV_7T_)/GTV_3T_.

**Table 3 diagnostics-13-01365-t003:** Intraclass correlation coefficient (ICC) for all the tumors and the different subgroups.

	ICC	Average Volume (cm^3^)	Mean Difference between Observers ± SD (cm^3^)	Mean Difference between Observers in % ± SD in %
All	0.969	24.1	2.74 ± 2.67	11.6 ± 6.92
T2	0.972	23.1	2.70 ± 2.72	12.4 ± 7.77
FLAIR	0.980	25.0	2.79 ± 2.71	10.8 ± 6.15
3T	0.975	24.9	2.81 ± 2.54	11.6 ± 6.66
7T	0.966	23.2	2.67 ± 2.89	11.6 ± 7.43
T2 (3T)	0.983	23.9	2.26 ± 1.77	11.0 ± 6.44
T2 (7T)	0.977	22.3	3.14 ± 3.53	13.7 ± 8.20
FLAIR (3T)	0.972	25.9	3.37 ± 3.18	12.1 ± 6.34
FLAIR (7T)	0.989	24.2	2.20 ± 2.25	9.5 ± 5.15

The difference between the observers is defined as (GTV_obs1_ − GTV_obs2_)/GTV_obs2_ and calculated in absolute values.

## Data Availability

The datasets used and/or analyzed during the current study are available from the corresponding author on reasonable request. The access to and use of the data must be in accordance with the rules of the Danish legislation and must be approved according to the Danish Data Protection Agency’s rules.

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
