# Peer review of "Delineation of Grade II and III Gliomas Investigated by 7T MRI: An Inter-Observer Pilot Study"

_diagnostics, 2023, doi:10.3390/diagnostics13081365_

Round 1
Reviewer 1 Report
The clinical benefits of 7T MRI scans remain unclear, but it has been suggested that there might be some improvement in tumor delineation using the stronger magnets. The overall conclusion of this study is that a good estimation of tumor volume can be obtained with both 3T and 7T MRI scans when imaging grade 2 and 3 gliomas.
Author Response
Reviewer 1
The clinical benefits of 7T MRI scans remain unclear, but it has been suggested that there might be some improvement in tumor delineation using the stronger magnets. The overall conclusion of this study is that a good estimation of tumor volume can be obtained with both 3T and 7T MRI scans when imaging grade 2 and 3 gliomas.
Reply:
Thank for you for your comment.
Reviewer 2 Report
The paper describes volume estimations of grade II and III brain tumours from 3T and 7T MR images in a very small patient cohort. A total of 7 patients, 6 of which had biopsy-verified grading, were scanned for T2-weigthed and T2-FLAIR at 3T and 7T within a month apart. Two raters delineated tumour boundaries for volume estimations from the images. The results indicate that both T2 contrast images from both fields provided comparable tumour volumes. The overall conclusion from the data set was that both 3T and 7T MR images allow tumour volume estimation with comparable results.
Specific points
1 The field strength and MRI images. The paper emphasises improved spatial resolution as the chief benefit of 7T. However, the facts that MRI contrast behaviours are strongly field dependent is overlooked. In fact, changing T1 and T2 relaxation rates from 3T to 7T are commonly exploited in tissue characterisation, including brain tumours, parallel to high spatial resolution at UHF.
2. MRI image quality. It is vital to provide numerical estimates from SNR and CNR from both types of images at both fields. Figure 1 indicates that chosen acquisition parameters have not produced optimal contrast at both fields. Along these lines, how were tumour boundaries defined? How many times were tumour volumes estimated from each case by each rater? What was the precision of volume estimates?
3. The patient cohort is very small, given that 6 of 7 patients had biopsy-verified diagnosis, the study must remain a pilot. The text must be adopted to this fact.
4. The cited literature is incomplete, several recent original and review articles dealing with brain tumour imaging at UHF are outstanding.
5. The paper is rather poorly written and will require substantial editing. There are several Tables and Figures from a small number of patients, number of both must be radically reduced to the essential.
6. Minor issues. In Abstract the order of 3T and 7T must be consistent, indicate pulse sequences in Abstract. Page 3, line 120, specify ‘0.7mm iso’, page 5, Table 1, what does ‘IDH-1’ refer to? mutation or not?
Author Response
Reviewer 2
The paper describes volume estimations of grade II and III brain tumours from 3T and 7T MR images in a very small patient cohort. A total of 7 patients, 6 of which had biopsy-verified grading, were scanned for T2-weigthed and T2-FLAIR at 3T and 7T within a month apart. Two ratters delineated tumour boundaries for volume estimations from the images. The results indicate that both T2 contrast images from both fields provided comparable tumour volumes. The overall conclusion from the data set was that both 3T and 7T MR images allow tumour volume estimation with comparable results.
Specific points
1 The field strength and MRI images. The paper emphasises improved spatial resolution as the chief benefit of 7T. However, the facts that MRI contrast behaviours are strongly field dependent is overlooked. In fact, changing T1 and T2 relaxation rates from 3T to 7T are commonly exploited in tissue characterisation, including brain tumours, parallel to high spatial resolution at UHF.
Reply:
Thank you for your comment. We agree. Water relaxation showed a highly significant increase (T1) and decrease (T2) with increasing field strength. However this does not directly involve our results, therefore we have not commented on it in the manuscript.
- MRI image quality. It is vital to provide numerical estimates from SNR and CNR from both types of images at both fields. Figure 1 indicates that chosen acquisition parameters have not produced optimal contrast at both fields. Along these lines, how were tumour boundaries defined? How many times were tumour volumes estimated from each case by each rater? What was the precision of volume estimates?
Reply:
SNR and CNR have been calculated and are stated on page 4 line 111. Tumour borders were defined visually. This is stated on page 5 line 124. The tumour volumes were estimated once by each delineator. This has been on page 5 line 125.
- The patient cohort is very small, given that 6 of 7 patients had biopsy-verified diagnosis, the study must remain a pilot. The text must be adopted to this fact.
Reply:
Wi have added “pilot study” both in the end of the introduction and in the conclusion.
- The cited literature is incomplete, several recent original and review articles dealing with brain tumour imaging at UHF are outstanding.
Reply:
We have added 2 references in the end of the introduction.
- The paper is rather poorly written and will require substantial editing. There are several Tables and Figures from a small number of patients, number of both must be radically reduced to the essential.
Reply:
We have carefully revised the manuscript. The old table 1 has been deleted. One table and two figures have been moved to supplementary material.
- Minor issues. In Abstract the order of 3T and 7T must be consistent, indicate pulse sequences in Abstract. Page 3, line 120, specify ‘0.7mm iso’, page 5, Table 1, what does ‘IDH-1’ refer to? mutation or not?
Reply:
Both items have been corrected.
Round 2
Reviewer 2 Report
There are several concerns remaining in the revised text as follows:
1. The title of the paper must incorporate ‘a pilot study’
2. Page 2, lines 53-56. Line 53, replace ‘ultra-high field’ by ‘7T’. It is unclear in regard to 7T MR scans, what quality in images improved detection of adenomas. Contrast, intratumour features or what?
3. Page 4, lines 115-20. Why are SNR and CNR values given without SD? This is rather unorthodox. How come 7T FLAIR SNRs be an order of magnitude greater than those of 3T?
4. Figure 1. Why are the single axial images windowed so differently. This concerns evidently also CSF, which is rather dark in 7T image compared to 3T. It is vital to window the images so that contrast between tumour and brain can be judged by readers. Also, why is only a GRIII case shown?
5. Tables 1 and 2. It would be vital to have mean with SD for GTV for both observers.
6. Figure 2 is redundant, Tables 1 and 2 show the essentials for GTV.
7. Discussion section is far too verbose, condense so that essential observations are covered in the light of relevant literature.
8. Conclusion section, page 17. The first sentence is rather bold about ‘clinical benefits of 7T’. A wealth of literature from several large clinics with decade’s of experience in 7T MRI argue otherwise.
Author Response
Comments and Suggestions for Authors
There are several concerns remaining in the revised text as follows:
- The title of the paper must incorporate ‘a pilot study’
Reply:
Pilot study has been incorporated in the title
- Page 2, lines 53-56. Line 53, replace ‘ultra-high field’ by ‘7T’. It is unclear in regard to 7T MR scans, what quality in images improved detection of adenomas. Contrast, intratumour features or what?
Reply:
Has been replaced by 7T
- Page 4, lines 115-20. Why are SNR and CNR values given without SD? This is rather unorthodox. How come 7T FLAIR SNRs be an order of magnitude greater than those of 3T?
Reply:
The SD has been added. We consulted the neuroradiologist and redid the measure of the reference areas for 7T FLAIR which changed the SNR calculation. The CNR is unchanged.
- Figure 1. Why are the single axial images windowed so differently. This concerns evidently also CSF, which is rather dark in 7T image compared to 3T. It is vital to window the images so that contrast between tumour and brain can be judged by readers. Also, why is only a GRIII case shown?
Reply:
We have commented on this in the part 4.3. This part has not been altered. We have only shown one delineation at 3T and 7T because it well illustrates the point we try to make. We regard showing more figures would not add any valuable information.
- Tables 1 and 2. It would be vital to have mean with SD for GTV for both observers.
Reply:
As stated in the manuscript each observer read once. Therefore, there is no SD.
- Figure 2 is redundant, Tables 1 and 2 show the essentials for GTV.
Reply:
Figure 2 shows the key-findings. Therefore, we have preferred to keep it and instead delete figure 3 and 4 in the supplementary material.
- Discussion section is far too verbose, condense so that essential observations are covered in the light of relevant literature.
Reply:
Superfluous parts of the discussion have been removed.
- Conclusion section, page 17. The first sentence is rather bold about ‘clinical benefits of 7T’. A wealth of literature from several large clinics with decades of experience in 7T MRI argue otherwise.
Reply:
Has been revised.
Round 3
Reviewer 2 Report
The concerns addressed